# Building Neural Networks that are Robust to Adversarial Examples Using Probabilistic Loss Function

## Abstract

Adversarial examples are an Achilles heel of deep neural networks, robbing them of their functional performance in mission-critical applications. This work proposes a novel method of making deep neural networks robust to adversarial training by modifying the loss function to a soft version of cross-entropy loss for classification problems. For regression problems, the data distribution is examined using Bayesian techniques (Gaussian Mixture Model) and the loss function is modified using a posterior probability distribution. The approach is justified using mathematical derivation and is supplemented by applying it on MNIST and Imagenet classification problems to demonstrate its robustness to FGSM and Carlinii-Wagner L2 attacks. This approach alleviates the overhead of training an additional model inherent in adversarial-distillation based methods.

## 1 Introduction

Deep neural networks have shown tremendous potential across a wide range of applications from image recognition to prediction in part because of their ability to generalize across applications with limited dependence on customized feature design and network architecture selection. This virtue of generalization makes them a tool of choice is some of the most challenging engineering tasks such as image analysis for threat detection and in self driving cars. Safety-critical applications such as deployment in self-driving vehicles also demand a high recall in addition to high precision. For example, detection of static or kinetic obstacles must show a high recall in addition to accurate perception of street direction (precision). Due to the lack of precise mathematical characterization of a deep neural network's performance, we must rely on test dataset to assure ourselves about its ongoing performance.

Adversarial examples are able to beguile deep neural networks into misclassifying the input to an incorrect class in the context of classification problems (Szegedy et al., 2013), (Goodfellow et al., 2015), (Papernot et al., 2016a). Such examples comprise of small perturbations to input and may be imperceptible to a human observer or may appear as easily distinguishable noise. The existence of adversarial examples casts doubts on the predictive power of deep neural networks in performance critical applications and reveals potential over-fitting in the training of deep neural networks. It illustrates the classic bias-variance trade-off – while increasingly complex models are necessary for tackling complex problems and reducing bias in training dataset, the models may be learning noise that is not repeatable in test dataset leading to increased variance.

Many solutions have been advanced by researchers to render neural networks robust to adversarial examples with varying degrees of success. There are two distinct taxonomies for classifying defenses against adversarial examples. First taxonomy is based on the existence of provable guarantees against the existence of adversarial attacks. Such networks provide a certified defense (Raghunathan et al., 2018). A second classification is based on a more functional approach based on methods use to make DNN robust to adversarial attacks. According to this taxonomy, the defenses can be classified into one of the following categories.

1. Adversarial training: This approach involves generating adversarial examples during the training process and including them in the dataset to make the model more resilient. How-

ever, this approach increases the size of dataset, making the training more expensive in terms of computation time and memory (Zhao et al., 2024).

2. Regularization: This approach augments the loss function used during training with additional terms to guide the model's parameters towards solutions that are more resistant to adversarial perturbations by reducing over-fitting. For example, L2 regularization using model parameters can serve as a safeguard against over-fitting and make the trained network robust to adversarial attacks.

3. Defensive distillation: This method uses a second model that is trained using the raw outputs of the original model (Papernot et al., 2016b). The second model is simpler than the original model and is unable to fit the noise that the original model was able to fit, while learning the reproducible patterns, thus making the model less susceptible to adversarial attacks. The second model is used for making predictions. This approach incurs the overhead of training a second model.

4. Ensemble methods: These techniques combine predictions from multiple models to create a more robust predictor in the hope that the noise in the predictions of individual models corresponding to over-fitting will cancel out when averaged across models. This approach is more time consuming due to the requirement of training additional models.

5. Architecture modification: Some defenses modify the network's architecture directly to increase its robustness.

6. Input transformations: These methods pre-process the input data by reducing adversarial perturbations using averaging or filtering. The pre-processed inputs are used to train the model.

This work presents a new approach to rendering deep neural networks resistant to adversarial attacks by modifying the loss function in a manner that can be applied for both classification and regression problems. Unlike defensive distillation, it avoids the overhead of training two models. Unlike many data-centric approaches of defending against adversarial attacks, it does not need to augment the input data with additional adversarial examples, or to apply filtering or smoothing to pre-process input data. This translates to faster training relative to other defenses against adversarial attacks.

The remainder of this work is structured as follows. Section 2 lays out the mathematical groundwork for probabilistic loss functions and formulates those for classification and regression problems. Section 3 demonstrates the effectiveness of the defense against FGSM (Goodfellow et al., 2015) and Carlini-Wagner (Carlini & Wagner, 2017) attacks. Section 4 concludes the work.

## 2 PROBABILISTIC LOSS FUNCTION BASED DEFENSE

In order to motivate the reason for modifying the loss function for making a deep neural network robust to adversarial attacks, let us consider a classification problem. A deep neural network represented as $F(\mathbf{X})$ maps the input vector $X$ into one of the $N$ classes, as shown in equation 1.

$$F(\mathbf{X}) = \arg\max_k \hat{y_k} \equiv i$$
$$i \in 0, 1, \cdots, N-1$$
(1)

Typically, a DNN for classification has a softmax layer as the last layer that produces a probability for each of the $N$ classes, denoted by vector $\hat{\mathbf{y}}$. The class with the highest probability is the predicted class $i$. The loss function used in training the cross-entropy loss is shown in equation **??**. $\mathbf{y}$ is a one-hot vector with $N$ components. The component corresponding to the correct class is set to 1, with all others being 0.

$$L(\mathbf{y}, \hat{\mathbf{y}}) = -\sum_{k=0}^{N-1} y_k \log \hat{y_k} = -\log \hat{y_i}$$
$$y_i = 1$$
$$y_k = 0 \ \forall k \neq i$$
(2)

Adversarial examples are small perturbations added to the input $\mathbf{X}$ that are able to flip the predicted class from $i$ to $j$, as shown in equation 3. The distance metric used can be a vector norm, with L0, L2 and L$\infty$ norms commonly used in practice. In this work, L2 norm is used as the distance metric.

$$\left\| \mathbf{X} - \mathbf{X}^{'} \right\|_2^2 = \epsilon$$
$$F(\mathbf{X}) = i$$
$$F(\mathbf{X}^{'}) = j$$
(3)

Targeted attacks pick a particular class $j$ while untargeted attacks accept any class $j$ different from $i$.

From the above equations, one can observe that the reason adversarial attacks work is because the loss function puts all the weight on one class, using a one-hot vector as $\mathbf{y}$. If two classes are similar, the loss function forces the deep neural network to discern between two similar classes, often leading the network to learn noise and overfit to training data. As long as the network does not put the entire probability mass on the correct class, the loss function will not be minimized for the hypothetical case where we have just one training sample.

This observation leads to the hypothesis that putting a non-zero probability mass on all classes, with the correct class carrying the highest weight may reduce over-fitting and render the trained DNN less vulnerable to adversarial attacks. This intuition can be formalized in equations 4 and 5.

Equation 4 states that if we pick an input $\mathbf{X}^{'}$ different from $\mathbf{X}$ and move it closer to $\mathbf{X}$, the cross-entropy loss relative to the correct class $i$ should decrease and the sum of cross-entropy losses relative to all other classes should increase. $i$ is the ground-truth class for input $\mathbf{X}$.

$$\text{Reduce } \left\| \mathbf{X} - \mathbf{X}^{'} \right\|_2^2 \rightarrow L\left( f(\mathbf{X}^{'}), i \right) \text{ reduces}$$
$$\text{and } \sum_{k \neq i} L\left( f(\mathbf{X}^{'}), k \right) \text{ increases}$$
(4)

Likewise, equation 5 states that as $\mathbf{X}^{'}$ moves away from $\mathbf{X}$, the losses behave as shown. Cross-entropy loss only considers the loss relative to the correct class $i$ and pays no attention to the loss relative to other classes.

$$\text{Increase } \left\| \mathbf{X} - \mathbf{X}^{'} \right\|_2^2 \rightarrow L\left( f(\mathbf{X}^{'}), i \right) \text{ increases}$$
$$\text{and } \sum_{k \neq i} L\left( f(\mathbf{X}^{'}), k \right) \text{ reduces}$$
(5)

In order to remedy this blind-spot in cross-entropy loss function, let us consider another loss function (shown in equation 6)possessing the properties shown in equations 4 and 5.

$$\tilde{L}\left( f(\mathbf{X}, \mathbf{y}) = L\left( f(\mathbf{X}, i) - \beta \sum_{k \neq i} L\left( f(\mathbf{X}, k) \right) \right)$$
$$1 - (N - 1)\beta = 1$$
(6)

We can generalize this loss function so that it is applicable to regression problems, as shown in equation 7. This equation shows the loss function for one input $\mathbf{X}, \mathbf{y}$ with the prediction of DNN denoted as $\hat{\mathbf{y}}$. It shows that if we have an outlier, i.e. an input data for which $\mathbf{y}$ is very different from the outputs of other input data points, the loss function will accord less weight to the outlier, thereby reducing the tendency of deep neural networks endowed with an abundance of free parameters from

over-fitting to outliers. This occurs because $\mathbf{X}_{\text{outlier}}$ is very different from the sample mean – $\mu$ – relative to the sample variance $\sigma^2$. An enhancement to this function involves clustering the data points and computing cluster mean and cluster variances followed by computing the modified loss function shown in equation 8.

$$\tilde{L}\left(\mathbf{y}, \hat{\mathbf{y}}\right) = \|\mathbf{y} - \hat{\mathbf{y}}\|_2^2 \exp - \left( \frac{\|\mathbf{X} - \mu\|_2^2}{2\sigma^2} \right)$$

$$\mu = \frac{1}{M} \sum_{k=0}^{M-1} \mathbf{X_k} \text{ where M is number of observations} \tag{7}$$

$$\sigma = \sqrt{\frac{\sum_{k=0}^{M-1} \|\mathbf{X} - \mu\|_2^2}{M-1}}$$

Let us unpack the significance of various terms in the loss function for regression problems in equation 8 in order to understand the source of its robustness to adversarial attacks. We first use the input features of the training dataset to cluster them into one of $C$ clusters using a clustering algorithm like K-means. We then compute the loss function using the cluster-membership of each data point. This loss function prevents the deep neural network from over-fitting to outliers within each cluster.

Run k-means clustering on all data points$\mathbf{X}_i$ in training data set

$C$ denotes thhe number of clusters

$$\tilde{L}\left(\mathbf{y}, \hat{\mathbf{y}}\right) = \|\mathbf{y} - \hat{\mathbf{y}}\|_2^2 \exp \left( -\frac{\|\mathbf{X} - \mu_t\|_2^2}{2\sigma_t^2} \right) \tag{8}$$

If point $\mathbf{X} \in$ cluster t with mean $\mu_t$ and variance $\sigma_t^2$

## 2.1 BAYESIAN PROBABILISTIC VERSION OF LOSS FUNCTIONS

The loss functions for classification and regression outlined earlier in equations 6 and 8 are deterministic and there is a potential for adversarial attacks to exploit this weakness. In this subsection, we formulate Bayesian probabilistic versions of the loss function that are offer an additional layer of defense against adversarial attacks.

Gaussian mixture model (GMM) offers a way to assign soft probabilities of class composition based on posterior probability distribution as described in Ahlawat (2025). For classification problems, we can begin with a prior probability that assigns a high weight to the output class and low weight to all other classes. We can then use EM algorithm to fine-tune the class composition probabilities, class means and variances using the training data. This is described in algorithm 1 and shown in equation 9. After the convergence of EM algorithm, we obtain the posterior class distribution probabilities as $\tau_k$ for each class $k$ and use that in the cross-entropy function. The modified loss function refrains from putting the entire weight on the ground truth output and uses Bayesian posterior probability as the modified ground truth probability.

$N$ classes and $M$ samples of data

Consider $j$ sample that belongs to class $i$

Class membership probability $= 0.9$ for correct class and $\dfrac{1 - 0.9}{N - 1}$ for other classes, denoted as $\tau_j(i)$ for i data point

**E step**

$$\tau_j(i) \leftarrow \frac{\tau_j(i) \exp\left(-\frac{(\mathbf{X_i} - \mu_\mathbf{i})^2}{2\sigma_i^2}\right)}{\sum_{k=0}^{M-1} \tau_j(k) \exp\left(-\frac{(\mathbf{X_i} - \mu_\mathbf{k})^2}{2\sigma_k^2}\right)}$$

**M step**

$$\mu_k \leftarrow \frac{1}{M} \sum_{l=0}^{M-1} \tau_k(l) \mathbf{X}_j$$

$$\sigma_k^2 \leftarrow \frac{1}{M-1} \sum_{l=0}^{M-1} \tau_k(l) \left(\mathbf{X}_j - \mu_k\right)^2$$

After convergence

$$\tilde{L}\left(f(\mathbf{X}, \mathbf{y}\right) = -\sum_{l=0}^{N-1} \tau_l \log \hat{y}_l$$

$$(9)$$

---

**Algorithm 1** Calibrating Probabilistic Loss Function for Classification

---

**Require:** $M$ data points $(\mathbf{X}, y)$ comprising the testing dataset.
1: Set $\tau_j(i) = 0.9$ and $\tau_k(i) = \frac{1-0.9}{N-1}$ for all classes $k \neq j$ for each data point $i$ where $j$ corresponds to the ground truth class $y(i)$
2: **for** iter $= 1, 2, \cdots$, until convergence of EM algorithm **do**
3:    Update class composition probability, $\tau$ using equation 10.

$$\tau_j(i) \leftarrow \frac{\tau_j(i) \exp\left(-\frac{(\mathbf{X_i} - \mu_\mathbf{i})^2}{2\sigma_i^2}\right)}{\sum_{k=0}^{M-1} \tau_j(k) \exp\left(-\frac{(\mathbf{X_i} - \mu_\mathbf{k})^2}{2\sigma_k^2}\right)} \qquad (10)$$

4:    Update the class mean and variances using equations 11 and 12.

$$\mu_k \leftarrow \frac{1}{M} \sum_{l=0}^{M-1} \tau_k(l) \mathbf{X}_j \qquad (11)$$

$$\sigma_k^2 \leftarrow \frac{1}{M-1} \sum_{l=0}^{M-1} \tau_k(l) \left(\mathbf{X}_j - \mu_k\right)^2 \qquad (12)$$

5:    Compute the change is $\mu$, $\sigma^2$ and $\tau$ from last iteration and terminate when parameters stabilize.
6: **end for**
7: Compute the loss function using equation 13.

$$\tilde{L}\left(f(\mathbf{X}, \mathbf{y}\right) = -\sum_{l=0}^{N-1} \tau_l \log \hat{y}_l \qquad (13)$$

---

Following a similar approach, the probabilistic loss function for regression problems can be formulated as shown in algorithm 2. For regression problems, we do not know the number of clusters. Therefore, this becomes a hyper-parameter. Another notable difference between the loss function calibration for classification and regression problems is the selection of prior class composition

probabilities, $\tau$. In classification problems, we select the known ground-truth class as the class with highest prior probability. For regression problems, on the other hand, we need to start with a non-informative prior that sets the class composition probabilities for all clusters to be the same value, i.e. $\frac{1}{N}$.

---

**Algorithm 2** Calibrating Probabilistic Loss Function for Regression

---

**Require:** $M$ data points $(\mathbf{X}, y)$ comprising the testing dataset.
**Require:** $N$ representing the number of clusters.
 1: Set $\tau_k(i) = \frac{1}{M}$ for all classes $k$ for each data point $i$.
 2: **for** iter = 1, 2, $\cdots$, until convergence of EM algorithm **do**
 3:     Update class composition probability, $\tau$ using equation 10.
 4:     Update the class mean and variances using equations 11 and 12.
 5:     Compute the change is $\mu$, $\sigma^2$ and $\tau$ from last iteration and terminate when parameters stabilize.
 6: **end for**
 7: Compute the loss function using equation 14.

$$\tilde{L}(\mathbf{y}, \hat{\mathbf{y}}) = \|\mathbf{y} - \hat{\mathbf{y}}\|_2^2 \sum_{l=0}^{N-1} \tau_l \exp\left(-\frac{\|\mathbf{X} - \mu_l\|_2^2}{2\sigma_l^2}\right) \tag{14}$$

---

It is worthwhile to note that Carlini-Wagner attack performs an optimization to find an adversarial example. Likewise, the defense involves calibrating the loss function using iterations of EM algorithm.

## 3 Testing the Defense's Effectiveness

In this section, we demonstrate the effectiveness of the defense. Let us consider the MNIST classification problem (Lecun et al., 1998) and deploy a deep neural network based on inception CNN architecture (Szegedy et al., 2015), (Szegedy et al., 2016).

The deep neural network comprises of 5,984,810 free parameters. A summary of the model architecture is available upon request. A snippet of parameter count can be found in the appendix. The model is trained using AdamW optimizer (I. & F., 2024) with weight decay not accumulating with iterations in momentum and variance. This is a variant of Adam optimizer (Kingma & Ba, 2014). The model is trained for 5 epochs with a learning rate of 0.001.

First, the model is trained using the vanilla cross-entropy loss function. The model attains an accuracy of 99% on the training dataset comprising of 60,000 examples. This network can be attacked by FGSM. Using an $\epsilon$ of 0.45 for generating FGSM perturbation, the model incorrectly classifies an image of digit 7 1 as 2, as shown in figure 2.

If we calibrate a loss function using algorithm 1 and train the deep neural network using it, FGSM is not able to fool the trained network.

We obtain similar results for Carlini-Wagner attack and on Imagenet dataset. Imagenet dataset has 1000 classes and offers greater prospect of successful adversarial attack due to the multitude of classes.

## 4 Conclusion

This work demonstrates the effectiveness of using a posterior probability based loss function for rendering a deep neural network robust to adversarial attacks. In the context of classification problems, adversarial attacks succeed because the cross-entropy loss function used for training a deep neural network rewards the network for over-fitting in order to reduce the loss. Regularization does not alleviate the problem entirely because it attempts to reduce network weights (for L2 regularization), which impacts the ability of the network to learn new patterns. The problem with cross-entropy loss function is that it puts the entire weight on one class that corresponds with the ground truth. In practice, each class has a probability of being the true class. For example, if two classes are very similar,

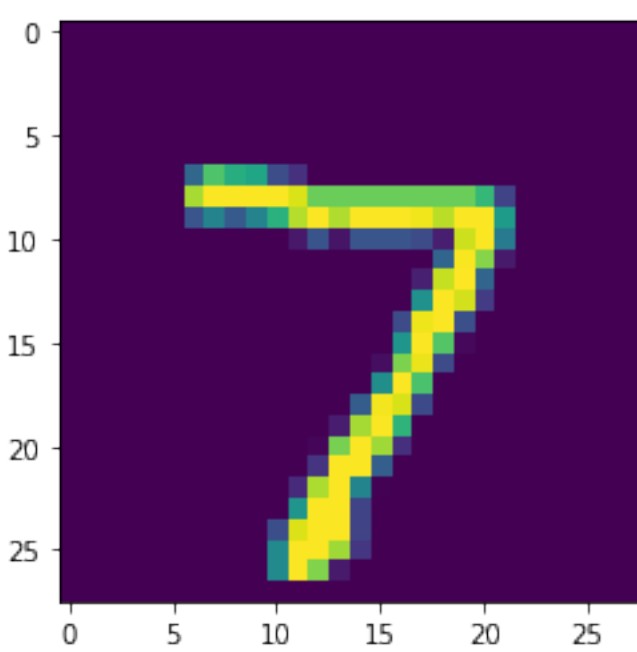

Figure 1: Example of Original 7 Image in MNIST Database

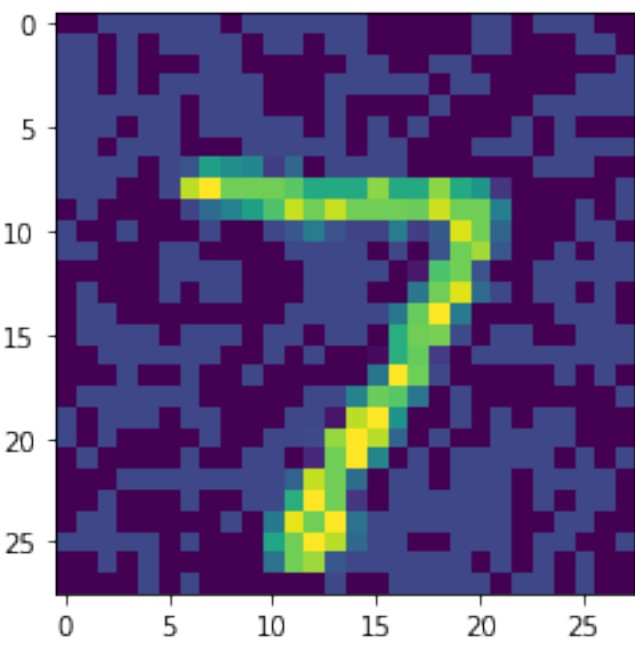

Figure 2: Using FGSM Attack Perturbed 7 Image Misclassified as 2

it will be hard to discern between them but the one-hot vector used in cross-entropy loss would arbitrarily assign the entire weight to one ground-truth class. This work uses Bayesian approach to fit a probability distribution to the training dataset using Gaussian mixture model. The resulting probability distribution is then used as the ground truth in cross-entropy loss. This approach can also

be extended to regression problems. Investigating the effectiveness of other non-Gaussian priors is an avenue for future research.

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

## A APPENDIX

Listing 1: Network Details for Classifying MNIST Dataset

```
================================================================
Total params: 5,984,810
Trainable params: 5,984,810
Non-trainable params: 0
Total mult-adds (M): 28.79
================================================================
Input size (MB): 0.00
Forward/backward pass size (MB): 0.93
Params size (MB): 23.94
Estimated Total Size (MB): 24.87
================================================================
```

