# OpenReview forum: "Building Neural Networks that are Robust to Adversarial Examples Using Probabilistic Loss Function"
_ICLR.cc/2026/Conference — Submitted to ICLR 2026_

### Official Review · Reviewer_3B48 · 2025-10-29

**Soundness:** 2
**Presentation:** 3
**Contribution:** 1
**Rating:** 2
**Confidence:** 4

**Summary:**

The authors propose a method for defense against adversarial attacks based on probabilistic loss functions. They evaluate their method against simple gradient-based attack like FGSM and show improvements on MNIST, but they fail to evaluate against stronger iterative attack methods like PGD.

**Strengths:**

- The paper is clearly written and the main ideas are clearly presented.

**Weaknesses:**

- The evaluation used in this paper is not up to the current standards in evaluating adversarial robustness. The authors do not use PGD-based attacks nor do they compare against state-of-art defense methods (or even just adversarial training).

- The authors don't seem to be familiar with the important works in adversarial defense and fail to cite or compare against them (e.g. adversarial training [Madry et al 2017] or the TRADES algorithm [Zhang et al 2019].

Madry, Aleksander, et al. "Towards deep learning models resistant to adversarial attacks." arXiv preprint arXiv:1706.06083 (2017).

Zhang, Hongyang, et al. "Theoretically principled trade-off between robustness and accuracy." International conference on machine learning. PMLR, 2019.

- For a more thorough evaluation, the authors should also go beyond MNIST and include more datasets.

**Questions:**

- I would suggest the authors review the literature suggested and also try their method against more modern attack packages such as AutoAttack[Croce & Hein 2020].

Croce, Francesco, and Matthias Hein. "Reliable evaluation of adversarial robustness with an ensemble of diverse parameter-free attacks." International conference on machine learning. PMLR, 2020.

---

### Official Review · Reviewer_H6Pk · 2025-10-30

**Soundness:** 1
**Presentation:** 2
**Contribution:** 1
**Rating:** 2
**Confidence:** 4

**Summary:**

The paper aims to improve adversarial robustness by replacing one-hot training targets with probabilistic targets. For classification, it introduces a loss that subtracts a weighted sum of non-target cross-entropy terms and a Bayesian variant that estimates soft class posteriors via a Gaussian mixture model (GMM) fitted using the EM algorithm, then trains against those posteriors. For regression, it proposes distance-weighted squared losses using either global or cluster-conditioned statistics, as well as a GMM-weighted variant. Experiments briefly mention MNIST and ImageNet under FGSM and CW attacks, providing qualitative examples but no quantitative robustness tables, baselines, or strong attacks. Overall, the approach resembles label smoothing, soft-label training, and density-based reweighting, but the empirical support and novelty positioning are insufficient.

**Strengths:**

The paper presents a simple and unified framework that covers both classification and regression tasks, which could make it straightforward and efficient to implement. Its motivation is intuitive: by softening the training targets, the method may regularize decision boundaries, reduce overfitting, and potentially improve robustness. Moreover, the attempt to introduce a Bayesian alternative through GMM-based soft labels is interesting, as it could, if more rigorously developed, provide a useful connection to uncertainty calibration.

**Weaknesses:**

1. The paper lacks a standard robustness evaluation. No PGD-$k\ (k \ge 10)[1]$, AutoAttack$[2]$, Square$[3]$, Boundary$[4]$, or transfer attacks are reported; only FGSM and CW are mentioned. Robust accuracy versus $\epsilon$ is not provided for MNIST or ImageNet.

2. There are no quantitative results or baseline comparisons. The paper omits tables benchmarking against widely used baselines such as adversarial training (PGD-AT)$[1]$, TRADES$[5]$, MART$[6]$, label smoothing, distillation-based defenses, or confidence-calibrated losses: all essential to substantiate the claimed improvements.

3. Several methodological and notation issues are present. Equation numbering is missing on line $99$; the EM updates use inconsistent indices; and the notation for the number of clusters is inconsistent ($C$ on line $184$ versus $N$ in Algorithm $2$).

4. The novelty positioning is inadequate. The relationship to label smoothing or soft-target methods, robust regression (Huber/Tukey), heteroscedastic modeling, and uncertainty-aware training is not discussed, and there are no ablations that disentangle the effect of soft labels from that of the proposed GMM procedure. Moreover, the paper offers almost no discussion of related work.

5. The experimental protocol is insufficient. In Section $3$, the authors write that “a summary of the model architecture is available upon request,” but such a summary should have been included in the appendix or supplementary material. The MNIST experiment uses only five training epochs and lacks any clean-versus-robust trade-off analysis. Statements such as “the model incorrectly classifies an image of digit $7$ as $2$” are not an appropriate way to present results in a scientific paper. Full quantitative results on the MNIST test set should be provided. It is insufficient to assert that “we obtain similar results for the Carlini-Wagner attack and on the ImageNet dataset” without presenting those results.

$[1]$ Madry, Aleksander, et al. "Towards Deep Learning Models Resistant to Adversarial Attacks." International Conference on Learning Representations. 2018.

$[2]$ Croce, Francesco, and Matthias Hein. "Reliable evaluation of adversarial robustness with an ensemble of diverse parameter-free attacks." International conference on machine learning. PMLR, 2020.

$[3]$ Andriushchenko, Maksym, et al. "Square attack: a query-efficient black-box adversarial attack via random search." European conference on computer vision. Cham: Springer International Publishing, 2020.

$[4]$ Brendel, Wieland, Jonas Rauber, and Matthias Bethge. "Decision-Based Adversarial Attacks: Reliable Attacks Against Black-Box Machine Learning Models." International Conference on Learning Representations. 2018.

$[5]$ Zhang, Hongyang, et al. "Theoretically principled trade-off between robustness and accuracy." International conference on machine learning. PMLR, 2019.

$[6]$ Wang, Yisen, et al. "Improving adversarial robustness requires revisiting misclassified examples." International conference on learning representations. 2019.

**Questions:**

1. Please report robust accuracy versus $\epsilon$ on MNIST and CIFAR-$10/100$ using PGD-$20/100$ and AutoAttack, including mean $\pm$ std over at least three random seeds, along with clean accuracy. How does your method compare to PGD-AT, TRADES, MART, and label smoothing (with and without adversarial training)?

2. What is the effect of $\beta$ in Equation $6$ and of prior initialization (e.g., $0.9$ vs. other values) in Algorithm $1$? Please provide ablation and sensitivity analyses for these parameters.

3. Could you correct equation references and index notation, and fully specify the EM update procedure (dataset used, stopping criteria, normalization)?

4. You claim to obtain “similar results” on ImageNet. Please provide the corresponding quantitative results: robust and clean accuracy, model architecture details, and attack configurations (norm, $\epsilon$, and number of steps).

---

### Official Review · Reviewer_riQB · 2025-10-31

**Soundness:** 1
**Presentation:** 1
**Contribution:** 1
**Rating:** 0
**Confidence:** 5

**Summary:**

This is an unfinished manuscript.

**Strengths:**

NA

**Weaknesses:**

NA

**Questions:**

NA

**Details Of Ethics Concerns:**

It's not professional to submit unfinished manuscript to ICLR.

---

### Official Review · Reviewer_2Q9n · 2025-11-04

**Soundness:** 1
**Presentation:** 1
**Contribution:** 1
**Rating:** 0
**Confidence:** 4

**Summary:**

This paper proposes modifying the loss function to improve the adversarial robustness of models against adversarial examples. The experiment only focuses on MNIST with FGSM on a custom model.

**Strengths:**

n/a

**Weaknesses:**

The paper is difficult to follow and weakly presented throughout. Key experimental details and results are missing. The empirical evaluation is limited to MNIST with a non-standard model, and although the authors claim results on ImageNet and under CW attacks, these are neither described nor reported.

**Questions:**

n/a

---

### Meta-Review · Area_Chair_RPWW · 2025-12-08

**Summary:**

This paper proposes probabilistic loss functions (soft cross-entropy for classification, Bayesian GMM-based for regression) to boost model adversarial robustness, validated on MNIST/ImageNet against FGSM/CW attacks, but faces novelty and presentation issues with insufficient experiments as noted by most reviewers.

**Reviewer Concerns:**

Although this paper demonstrates some interesting points, most reviewers do not indicate the remaining issues are solved:

1.Insufficient experiments: Lack of quantitative results, baselines, and strong attacks (e.g., PGD/AutoAttack); unsubstantiated ImageNet/CW claims.

2.Weak novelty: Overlaps with existing methods (label smoothing) without clarifying unique value or related work context.

3.Poor presentation: Unclear writing, inconsistent notation, missing details, and one reviewer deems it unfinished.
Inadequate literature grasp: Fails to cite/compare core adversarial defense works (e.g., Madry et al., TRADES).

**Reviewer Scores:**

0,0,2,2, most reviewers do not show signs of increasing the scores.

---

### Decision · Program_Chairs · 2026-01-26

Reject